# Automated construction of $U(1)$-invariant matrix-product operators from graph representations

**Sebastian Paeckel⋆, Thomas Köhler and Salvatore R. Manmana**

Institut für Theoretische Physik, Universität Göttingen, 37077 Göttingen, Germany

⋆ sebastian.paeckel@theorie.physik.uni-goettingen.de

## Abstract

We present an algorithmic construction scheme for matrix-product-operator (MPO) representations of arbitrary $U(1)$-invariant operators whenever there is an expression of the local structure in terms of a finite-states machine (FSM). Given a set of local operators as building blocks, the method automatizes two major steps when constructing a $U(1)$-invariant MPO representation: (i) the bookkeeping of auxiliary bond-index shifts arising from the application of operators changing the local quantum numbers and (ii) the appearance of phase factors due to particular commutation rules. The automatization is achieved by post-processing the operator strings generated by the FSM. Consequently, MPO representations of various types of $U(1)$-invariant operators can be constructed generically in MPS algorithms reducing the necessity of expensive MPO arithmetics. This is demonstrated by generating arbitrary products of operators in terms of FSM, from which we obtain exact MPO representations for the variance of the Hamiltonian of a $S = 1$ Heisenberg chain.



# 1 Introduction

The density-matrix renormalization group (DMRG) [1–5] has proven to be one of the most powerful tools to treat low-dimensional problems in strongly correlated quantum systems. Inspired by quantum information theory, a formulation in terms of matrix-product states (MPS) [6] and matrix-product operators (MPO) [5] boosted the development of related algorithms (e.g., [7–10]). Numerous different techniques have been developed in the framework of MPS, most of which require the representation of the Hamiltonian of the system and of further observables in terms of MPOs [11–13]. Even though finding such a representation is a well-known and generally solved problem [14–16], it can become arbitrarily complicated as systems may incorporate complex lattice geometries, site-dependent interaction strengths, or transformations of local operators due to their commutation relations.

It is possible to build generic construction schemes for MPO representations of operators by means of finite-state machines (FSM) [15]. However, the numerical realization of these FSMs can be quite involved, especially when exploiting global symmetries [14, 17, 18].

In this paper, we use the fact that FSMs have an underlying graph structure to obtain a generic algorithmic construction scheme for MPO representations of operators conserving $U(1)$ symmetries. Consequently, most of the complexity can be unwrapped into tensor-network manipulations of strings of local operators, which can be mapped one-to-one to graph representations. The obtained construction scheme can be automatized so that an implementation is capable of efficiently creating MPO representations. In particular, this also allows us to include commutation rules as well as conservation laws. We demonstrate how to map the operator arithmetics to operations on graph representations, leading to an algorithm for computing the sum as well as the product of two arbitrary operators. We use this algorithm to demonstrate how to compute the variance of a Hamiltonian $var\left(\hat{H}\right) = \left\langle \left(\hat{H} - \left\langle \hat{H}\right\rangle\right)^2 \right\rangle$ with very high accuracy. This is achieved by reducing the effects of catastrophic cancellation [19] and the total amount of required floating-point operations by using exact operator arithmetics in terms of FSM graphs. In addition, this also minimizes the overall computational costs.

# 2 $U(1)$-invariant tensor networks and application of two-site gates

Consider an operator $\hat{O}$ acting on a Hilbert space $\mathcal{H}$, which decomposes into a tensor product of $L$ identical $d$-dimensional local Hilbert spaces $\mathcal{H}_d$ ($d, L \in \mathbb{N}$),

$$\mathcal{H} = \mathcal{H}_d^{\otimes L} \tag{1}$$

$$\hat{O} : \mathcal{H} \longrightarrow \mathcal{H}, \tag{2}$$

and assume that this operator is invariant under a global $U(1)$ symmetry generated by local observables $\hat{n}_i, i \in \{1, \dots, L\}$,[1]

$$\left[\hat{N}, \hat{O}\right] = 0, \quad \hat{N} = \sum_{i=1}^{L} \hat{n}_i. \tag{3}$$

---

[1] Note that by $\hat{n}_i$ we denote the local generators of the symmetry and not the local density operators, which play this role only in the case of conservation of the total particle number.

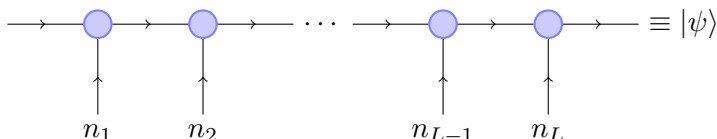

Figure 1: MPS representation of an arbitrary $U(1)$-invariant quantum state $|\psi\rangle$. The circles represent site tensors with physical degrees of freedom labeled by $n_i$. Edges attached to circles correspond to the indices of the site tensors $T^{n_i} \equiv T^{n_i}_{\alpha_{i-1},\alpha_i}$. Connected edges represent index contractions.

A state $|\psi\rangle \in \mathcal{H}$ can be expanded into an MPS with local basis states $\{|n_i\rangle\}$ spanned by the generators $\{\hat{n}_i\}$ of the global $U(1)$ symmetry

$$|\psi\rangle = \sum_{n_1 \cdots n_L} \langle n_1 \cdots n_L | \psi \rangle |n_1 \cdots n_L\rangle \equiv \sum_{n_1 \cdots n_L} T^{n_1} \cdots T^{n_L} |n_1 \cdots n_L\rangle , \qquad (4)$$

with $T^{n_i} \equiv T^{n_i}_{\alpha_{i-1}\alpha_i}$ being rank-3 tensors acting on site $i$, which carry a physical bond index $n_i$ and two virtual bond indices $\alpha_{i-1}, \alpha_i$. The coefficients $\langle n_1 \cdots n_L | \psi \rangle$ can be reobtained by contracting the corresponding tensors $T^{n_1} \cdots T^{n_L}$ over all their virtual indices. In a graphical notation, a tensor $T$ is represented by drawing a shape (circles, squares, ...) with as many legs attached to it as there are tensor indices (see fig. 1). Contractions over indices are denoted by connecting the corresponding edges. This yields a tensor network.

Due to conservation of the global quantum number $\hat{N}$, these rank-3 site tensors $T^{n_i}$ are irreducible representations of the global $U(1)$ symmetry. The physical indices $n_i$ label the basis states of the local generators. As a consequence, the site tensors can be decomposed into blocks that are subject to an on-site conservation law [17, 18]

$$T^{n_i} \stackrel{\sim}{=} T^{n_i}_{\alpha_{i-1},\alpha_i} \delta(\alpha_{i-1} + n_i - \alpha_i), \qquad (5)$$

with the block indices $(n_i, \alpha_{i-1}, \alpha_i)$ labeling the irreducible representations of the global quantum number $\hat{N}$ on the particular site $i$. These tensor blocks in general are complex matrices $T^{n_i}_{\alpha_{i-1},\alpha_i} \in \mathbb{C}^{d_{\alpha_{i-1}} \times d_{\alpha_i}}$ of dimensions $d_{\alpha_{i-1}} \times d_{\alpha_i}$.[2] Hence, thinking in terms of block-diagonal MPS, each site matrix $M^{n_i}$ is decomposed into the block diagonal form $M^{n_i} = \bigoplus_a T^{n_i}_a$ with the non-vanishing blocks $T^{n_i}_a = T^{n_i}_{\alpha_{i-1},\alpha_i}$ having block dimensions $d_{\alpha_{i-1}} \times d_{\alpha_i}$ and $a$ labeling the irreducible representations at site $i$.

There is a simple graphical representation for these local conservation laws acting on the tensor blocks, which is derived from the tensor-network framework [17]. Whenever there are indices with a (hidden) plus sign in the $\delta$ function, the corresponding bonds in the network are labeled by an ingoing arrow, whereas, for indices with a minus sign, an outgoing arrow is placed on the respective bond (see fig. 1).

Next, we take a more elaborate look at the action of a two-site gate on the symmetry conserving state, ignoring the common framework of irreducible representations and block diagonal structures for a moment (later we generalize the considerations to arbitrary operator expressions). Introducing operators $\hat{A}^{(i)}, \hat{B}^{(j)} : \mathcal{H}^{(i,j)}_d \longrightarrow \mathcal{H}^{(i,j)}_d$ acting only on the sites $i, j$, we start from a generic $U(1)$-invariant expression of the form

$$\hat{A}^{(i)}\hat{B}^{(j)}|\psi\rangle \equiv \sum_{n_1 \cdots n_L} \sum_{n'_i,n'_j} T^{n_1} \cdots A^{n_i n'_i} T^{n_i} \cdots B^{n_j n'_j} T^{n_j} \cdots T^{n_L}$$

$$\times \delta(n_i + n_j - (n'_i + n'_j)) |n_1 \cdots n_L\rangle , \qquad (6)$$

---

[2]For brevity, in the following, we suppress bond indices whenever it is clear, which matrix contractions have to be performed, or if the contractions themselves are irrelevant for the discussion.

where the $\delta$ function ensures that the total quantum number $\hat{N}$ is conserved when applying the operator. A dummy index $\tau$ with $-(d-1) \leq \tau \leq (d-1)$ is introduced to factorize the $\delta$ function. Thus, suppressing the sums over the physical indices $n_i$ for a moment, we obtain

$$\hat{A}^{(i)}\hat{B}^{(j)}|\psi\rangle = \sum_{n_i',n_j'}\left\{\cdots T^{n_{i-1}}\left[\sum_{\tau} \hat{A}^{n_i n_i'}T^{n_i}\delta(\tau-(n_i-n_i'))T^{n_{i+1}}\cdots\right.\right.$$
$$\left.\left.\times \hat{B}^{n_j n_j'}T^{n_j}\delta(\tau-(n_j'-n_j))\right]T^{n_{j+1}}\cdots|n_1\cdots n_L\rangle\right\}, \quad (7)$$

which nearly restores the factorized shape of generic MPS, even though the physical sites $i, j$ are still connected by the sum over the dummy index $\tau$. It is desirable to restore a tensor-network form so that the action of an operator pair can be written as contraction of tensors acting on the local Hilbert spaces. Therefore, we need to take a closer look at the matrix elements $A^{n_i n_i'}_{\gamma_{i-1}\gamma_i}, B^{n_j n_j'}_{\gamma_{j-1}\gamma_j}$ of the local operators $\hat{A}^{(i)}, \hat{B}^{(j)}$. If the total operator expression is $U(1)$-invariant, then the local operators themselves have to be one-dimensional representations in the physical indices (acting on the local Hilbert space) so that each operator carries a unique mapping between states $|n_i\rangle \rightarrow |n_i'\rangle$. In other words, for $U(1)$-invariant operator pairs each local operator is non-vanishing only for a certain value of the dummy index $\tau' = \Delta$. For instance, in case of spin-ladder operators $(\hat{S}^{\pm})^{(i)}$ the total value of $S^z$ is locally changed by $\pm 1$. Hence,

$$(\hat{S}^{\pm})^{n_i, n_i'} \neq 0 \Leftrightarrow n_i - n_i' = \pm 1 \equiv \Delta, \quad (8)$$

where we introduced the change of local quantum numbers $\Delta$. Having this in mind, block-index conservation laws for the local operators can be realized via

$$\hat{A}^{(i)} \equiv A^{n_i n_i'}_{\tau_{i-1}\tau_i}\delta(\Delta-(n_i-n_i'))\delta\left((n_i+\tau_{i-1})-(n_i'+\tau_i)\right) \quad (9)$$

$$\hat{B}^{(j)} \equiv B^{n_j n_j'}_{\tau_{j-1}\tau_j}\delta(\Delta-(n_j'-n_j))\delta\left((n_j+\tau_{j-1})-(n_j'+\tau_j)\right). \quad (10)$$

The case of two-site gates is obtained by setting $\tau_{i-1} = \tau_j \equiv 0$ as well as $\tau_i = \tau_{j-1} \equiv \tau$; the non-vanishing operator blocks $A^{n_i n_i'}_{\tau_{i-1}\tau_i}, B^{n_j n_j'}_{\tau_{j-1}\tau_j} \in \mathbb{C}^{d_{\tau_{i-1,j-1}} \times d_{\tau_{i,j}}}$ now contain the reduced operator matrix elements acting on the local basis states $|n_{i,j}'\rangle$. The contraction of the block index $\tau$ over the intermediate site tensors $T^{n_k}, i < k < j$ can be recast into a matrix contraction using the matrix identity

$$(A_1, \cdots, A_n)\begin{pmatrix} D & \cdots & 0 \\ \vdots & \ddots & \vdots \\ 0 & \cdots & D \end{pmatrix}\begin{pmatrix} B_1 \\ \vdots \\ B_n \end{pmatrix} = \sum_{i=1}^{n} A_i D B_i. \quad (11)$$

Therefore, the application of the above two-site gate is factorized completely if we define intermediate shift tensors that act on the sites $k$ with $i < k < j$ as identity, but are contracted over auxiliary bond indices $\tau_k$,

$$\hat{R}^{(i)}_{\Delta} = R^{n_i n_i'}_{\tau_{i-1}\tau_k}(\Delta)\delta((n_i+\tau_{i-1})-(n_i'+\tau_i)) \equiv \delta(n_i-n_i')\delta(\tau_{i-1}-\Delta)\delta(\Delta-\tau_i). \quad (12)$$

Hence, the action of the $U(1)$-invariant operator pair can conveniently be written as tensor network,

$$\hat{A}^{(i)}\hat{B}^{(j)}|\psi\rangle = \sum_{n_1\cdots n_L} T^{n_1}\cdots T^{n_{i-1}}V^{n_i}P^{n_{i+1}}\cdots P^{n_{i-1}}W^{n_j}T^{n_{j+1}}\cdots T^{n_L}|n_1\cdots n_L\rangle \quad (13)$$

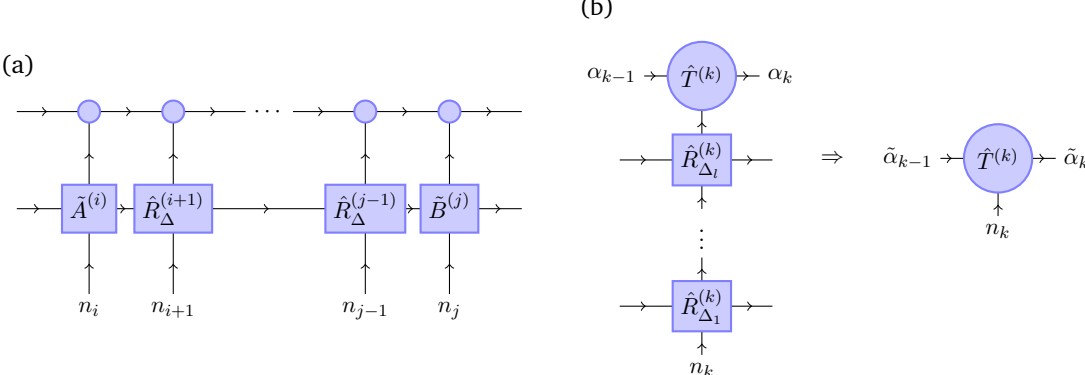

Figure 2: (a) MPO representation of the $U(1)$-invariant operator expression $\hat{A}^{(i)}\hat{B}^{(j)}$ with shift tensors $\hat{R}_\Delta^{(k)}$ mediating the shift in the auxiliary-bond quantum numbers. (b) Network of various shift tensors applied to the $U(1)$-invariant site tensor $\hat{T}^{(k)}$ before and after performing the contractions, respectively. After contraction, the auxiliary block indices are given via $\tilde{\alpha}_{k-1,k} = \alpha_{k-1,k} + \Delta_1 + \cdots + \Delta_l$.

with the definitions

$$V^{n_i} = \sum_{n_i'} A^{n_i n_i'}_{\tau_{i-1}\tau_i} \delta(\Delta - (n_i - n_i'))\delta((n_i + \tau_{i-1}) - (n_i' + \tau_i))T^{n_i'} \tag{14}$$

$$P^{n_k} = \sum_{n_k'} R^{n_k n_k'}_{\tau_{k-1}\tau_k}(\Delta)\delta((n_k + \tau_{k-1}) - (n_k' + \tau_k))T^{n_k'} \tag{15}$$

$$W^{n_j} = \sum_{n_j'} B^{n_j n_j'}_{\tau_{j-1}\tau_j} \delta(\Delta - (n_j' - n_j))\delta((n_j + \tau_{j-1}) - (n_j' + \tau_j))T^{n_j'} . \tag{16}$$

When further local operators to the left and right of $\hat{A}^{(i)}, \hat{B}^{(i)}$ are absent, the auxiliary indices are shrunk to dummy indices $\tau_{i-1} = \tau_j \equiv 0$. Thus, the sum over all remaining auxiliary block indices $\tau_{i \leq l \leq j}$ restores the global conservation law. Note that we have implicitly fixed a gauge freedom carried by the auxiliary indices $\tau_i$ to $\tau_0 = \tau_L \equiv 0$ (we can even go further and permit any global change $\delta$ of the overall quantum number $\hat{N}$ by setting $\tau_L = \delta$).

Even though these expressions look a bit tedious, they can be represented compactly in form of a tensor network, which also reveals how useful this decoupling turns out to be (see fig. 2a). For example, it is possible to identify the well-known transformation law for $U(1)$-invariant MPO site tensors in the expressions above [17], yet the block indices $\tau_k$ are related to the change of the quantum number, captured by the shift tensors $R_\Delta^{(k)}$.

Generally, contractions over physical bond indices $n_k, n_k'$ of such shift tensors correspond to a mapping from one block $a$ of an irreducible representation $U^{(k)}(N) = \bigoplus_a U_a^{(k)}(N)$ on a site $k$ to another block $a + \Delta$. Hence, given a decomposition into the different quantum-number sectors of an $U(1)$-invariant MPS or MPO tensor

$$\hat{T}^{(k)} = \bigoplus_a \hat{T}_a^{(k)} , \tag{17}$$

the contraction along a chain of $l$ shift tensors over physical indices (with $\Delta_1 \ldots \Delta_l$ the changes in the local quantum numbers) is given via

$$\hat{T}^{(k)}_{a+\Delta_1+\cdots+\Delta_l} = \hat{R}_{\Delta_1}^{(k)} \ldots \hat{R}_{\Delta_l}^{(k)} \hat{T}_a^{(k)} . \tag{18}$$

From this point on it is clear how to generalize the considerations to arbitrary strings of local operators. In a given expression of local operators, we need to identify pairs of operators

that conserve the global $U(1)$ quantum number but locally change the on-site quantum numbers $n'_{i_1} \to n_{i_1}, n'_{i_2} \to n_{i_2}$. For each pair, shift tensors then have to be inserted acting on the intermediate sites $i_1 < k < i_2$. Note that in a code there is no need to explicitly implement the shift tensors. It suffices to implement only their action on the virtual bonds of either the MPS or of another MPO, i.e., to collect all vertical strings of shift tensors $\left\{ \hat{R}^{(k)}_{\Delta_r} \right\}_{1 \leq r \leq l}$ in the network and to apply the shifting in the auxiliary block indices $\alpha_{k-1,k} \to \tilde{\alpha}_{k-1,k} = \alpha_{k-1,k} + \Delta_1 + \cdots + \Delta_l$ (see fig. 2b).

# 3 $U(1)$-invariant MPO representation from FSMs

As discussed in [15, 16], the MPO representation of an operator on a tensor-product space $\mathscr{H} = \mathscr{H}_d^{\otimes L}$ can be obtained from the transition amplitudes of FSMs. In the following, the underlying graph structure of FSMs is used extensively. Thus, we first give a brief review on how to identify FSMs with MPO representations, following [15].

## 3.1 MPO construction from FSMs

Let $\mathscr{K} = \left\{ \hat{o}_1^{(i)}, \cdots, \hat{o}_m^{(i)} \right\}$ be a set of $m \in \mathbb{N}$ *local* operators $\hat{o}_i : \mathscr{H}_d \to \mathscr{H}_d$. Any global operator is of the general form

$$\hat{H} = \sum_{\nu,r} \hat{H}_{\nu,r} = \sum_{\nu,r} \sum_i \hat{h}^{(i)}_{\nu,r} \, , \tag{19}$$

with $\hat{h}^{(i)}_{\nu,r} = f^r_{\nu_1 \dots \nu_n} \hat{o}^{(i)}_{\nu_1} \cdots \hat{o}^{(i+r)}_{\nu_n}$ being strings of $n$ local operators $\hat{o}_\nu \in \mathscr{K}$ coupling lattice sites $i$ with range $r + 1 \geq n$, amplitude $f^r_{\nu_1 \dots \nu_n} \in \mathbb{C}$, and abbreviating the index set $\nu = (\nu_1 \dots \nu_n)$. For later convenience we will call $\hat{h}^{(i)}_{\nu,r}$ *lattice ordered $n$-point $r + 1$-ranged operator strings*.

The set of all lattice-ordered $n$-point $r + 1$-ranged operator strings $\Sigma = \left\{ \hat{h}^{(i)}_{\nu,r} \right\}_{\nu,r}$ defines a language of a FSM, i.e., there is a set of states $\mathscr{L}$ and a transition function $\delta : \mathscr{L} \times \mathscr{K} \longrightarrow \mathscr{L}$ so that with a proper initial and final state $I, F \in \mathscr{L}$, the FSM $M(\mathscr{K}, \mathscr{L}, \delta, I, F)$ is obtained with a generated language $\Sigma$. The corresponding graph is then a representation of $\hat{H}$ and is denoted by $\Lambda(\hat{H})$.

An example for the graph representation of the XX model is given in fig. 3a and in the following, we shortly explain how to obtain this graph. At first, we have to rewrite the global operator in terms of lattice-ordered $n$-point $r + 1$-ranged operator strings[3]

$$\hat{H}_{XX} = \sum_i \hat{S}^+_{(i)} \hat{S}^-_{(i+1)} + \sum_i \hat{S}^-_{(i)} \hat{S}^+_{(i+1)} \equiv \sum_i \hat{h}^{+-}_{(i)} + \sum_i \hat{h}^{-+}_{(i)}, \tag{20}$$

i.e., we have two distinct 2-point 2-ranged operator strings. Then, we define a default set of states $\mathscr{L}_0 = \{I, F\}$ with $I, F$ being the initial and final state of the FSM, respectively. For convenience, we will highlight them in the corresponding graphs with green ($I$) and red ($F$) background. In general, FSMs are capable to generate sequences of symbols $\hat{o} \in \mathscr{K}$ by transitioning between states $a \in \mathscr{L}$ via permitted transitions, i.e., those with non-vanishing transition function $\delta(\hat{o}, a)$. Thus, we define the initial/final state by demanding that all the previous/subsequent transitions have to be identities.

Next, we build the graph fig. 3a by constructing each operator string separately, so we may start with $\hat{h}^{+-}_{(i)}$. We therefore add $r = 1$ intermediate states (here only one state $A_1 \equiv A$) to the set of states: $\mathscr{L} = \mathscr{L}_0 \cup \{A\}$ and define permitted transitions $\delta(I, \hat{S}^+) = A$ and $\delta(A, \hat{S}^-) = F$.

---

[3]In this example we change our convention and write site indices as lower indices.

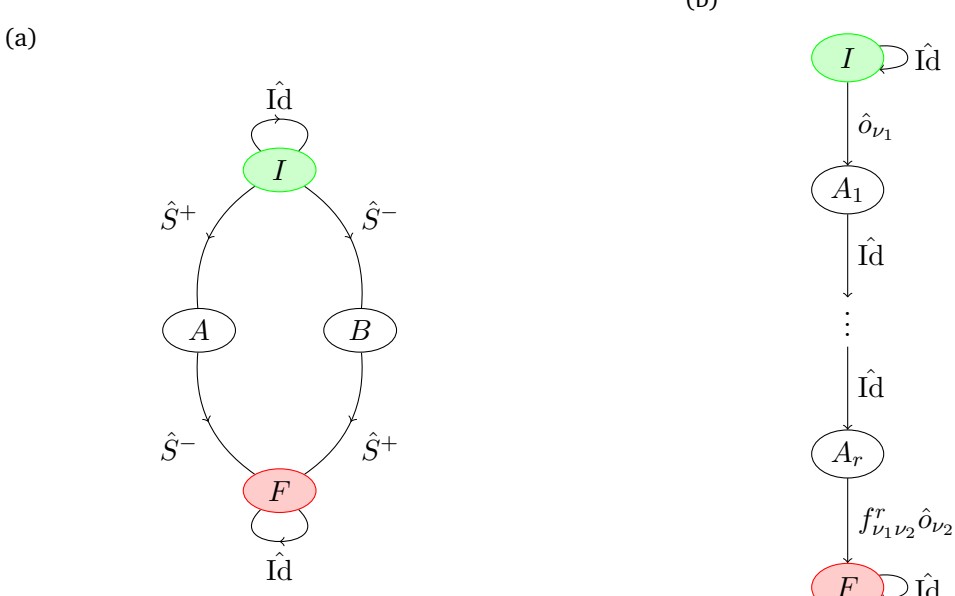

Figure 3: (a) Graph representation $\Lambda(\hat{H}_{XX})$ of the XX model $\hat{H}_{XX} = \sum_i \hat{S}^+_{(i)}\hat{S}^-_{(i+1)} + \text{h.c.}$. (b) Graph representation of an operator with only one 2-point interaction term $f^r_{\nu_1\nu_2}\hat{o}^{(i)}_{\nu_1}\hat{o}^{(i+r)}_{\nu_2}$ of range $r$. Note that the initial and final states are highlighted with a green and red background, respectively while intermediate states $A, B, A_1 \ldots A_r$ are left white.

Thus, the FSM can transition from the initial state (after placing an arbitrary number of identities) into state $A$ by appending an operator $\hat{S}^+$ onto the so far constructed operator string. Being in state $A$, the FSM has no other choice than to transition into state $F$ by appending a subsequent operator $\hat{S}^-$. A generalization of the construction of such local operator strings is shown in fig. 3b for the case of a general 2-point $r$-ranged operator string.

From this point on, we will call graphs that generate only one distinct type of $n$-point $r + 1$-ranged operator string *single-branched graphs* and identify them with the corresponding contribution in the global operator $\hat{H}_{\nu,r}$.

Returning to the XX model, we can finish the construction of the FSM by adding another single-branched graph transitioning from the initial state into an additional state $B$ via a local operator $\hat{S}^-$ and a transition into the final state via a local operator $\hat{S}^+$.

Finally, we can construct the MPO representation, i.e., the operator-valued matrices $\hat{W}^{n_i n'_i}$, by assigning matrices $W^{n_i n'_i}_{\tau_{i-1}\tau_i}$ of dimension $|\mathscr{L}| \times |\mathscr{L}|$ for all non-vanishing symmetry blocks $(n_i, n'_i, \tau_{i-1}, \tau_i)$ (i.e., those with at least one local operator block $\hat{o}^{n_i n'_i}_{\tau_{i-1}\tau_i} \neq 0$). Figure 4a sketches a general FSM and fig. 4b shows the generated MPO bulk tensor, in which the rows and columns are labeled by the states of the FSM. The corresponding boundary tensors are obtained by projecting out (a) the transition from the initial state into the bulk for $i = 1$ and (b) the transitions from the bulk into the final state for $i = L$. We emphasize that the site-dependent coefficients $c^{(i)}_{ab} \in \mathbb{C}$ are free parameters and therefore can be chosen independently for every site.

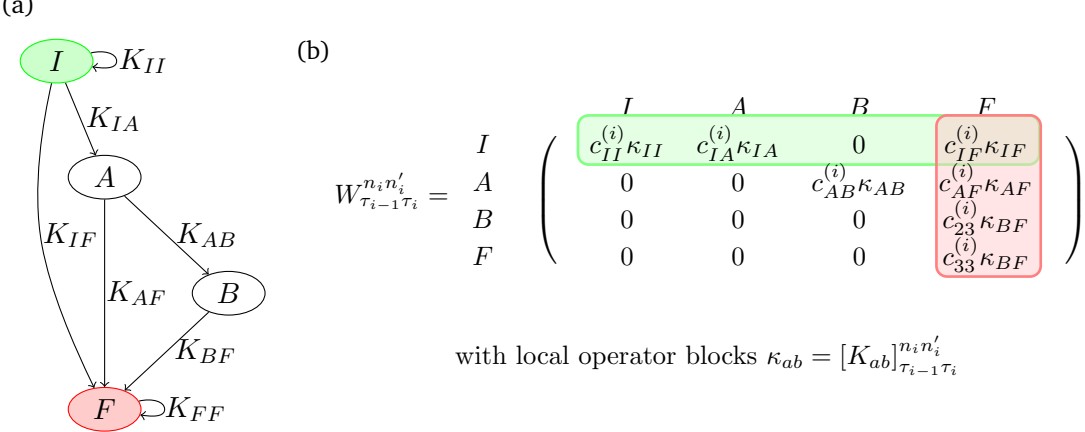

with local operator blocks $\kappa_{ab} = [K_{ab}]^{n_i n_i'}_{\tau_{i-1}\tau_i}$

Figure 4: (a) FSM defined on states $\mathscr{L} = \{I, A, B, F\}$ with transition amplitudes $K_{ab} \in \mathscr{K}$. The initial and the final state are highlighted in green or red, respectively. Transitions between states are denoted by arrows with the corresponding transition amplitudes $K_{ab}$. (b) Bulk MPO site-tensor block $W^{n_i n_i'}_{\tau_{i-1}\tau_i}$, obtained from the FSM in (a). The initial and the final site tensor are marked by a green or red background, respectively. The coefficients $c^{(i)}_{ab}$ are site-dependent weight functions, which can be used to introduce position-dependent transition amplitudes.

## 3.2 Maximally branched representation and local transformations on graphs

As already mentioned, every global operator on $\mathscr{H}$ can be formulated as a sum over lattice-ordered $n$-point $r + 1$-ranged operator strings

$$\hat{H} = \sum_{\nu, r} \hat{H}_{\nu, r}. \tag{21}$$

Note that the graph representation via FSM is not unique; for every operator $\hat{H}$, there is a set of corresponding FSMs $\{\Lambda(\hat{H})\}_\Lambda$. Therefore, we are free to choose one representation $\Lambda(\hat{H})$, which makes it easier to perform operator arithmetics and then switch to another representation $\tilde{\Lambda}(\hat{H})$ to find the most compact MPO.

Referring to eq. (21), a natural translation to the graph representations of $\hat{H}$ can be obtained by introducing a commutative map $\oplus$ between graph representations of sums of operators $\hat{H}_1, \hat{H}_2$ via

$$\oplus: \quad \Lambda(\hat{H}_1 + \hat{H}_2) = \Lambda(\hat{H}_1) \oplus \Lambda(\hat{H}_2). \tag{22}$$

The realization of $\oplus$ in terms of graphs is obtained by taking the graph representations of the operators $\hat{H}_1, \hat{H}_2$ and by merging the initial and final states as depicted in fig. 5.

Next, we can define the notion of a maximally branched graph representation, which is given by the graph $\Lambda_{\max}(\hat{H})$ satisfying the conditions: a) the initial state $I$ is the only state with more than one child state, b) the final state $F$ is the only state with more than one parent state. $\Lambda_{\max}(\hat{H})$ satisfies the equation

$$\Lambda_{\max}(\hat{H}) = \bigoplus_{\nu, r} \Lambda(\hat{H}_{\nu, r}). \tag{23}$$

This representation has several advantages; most importantly for our discussion, any local transformation of the operator can be mapped one-to-one to the transitions of the graph representation. Here local transformations are those transformations that map a lattice-ordered $n$-point $r + 1$-ranged operator string into another lattice-ordered $m$-point $r + 1$-ranged operator string $\hat{H}_{\nu, r} \to \hat{\tilde{H}}_{\tilde{\nu}, r}$ without changing $r$. To clarify what is meant by this mapping we

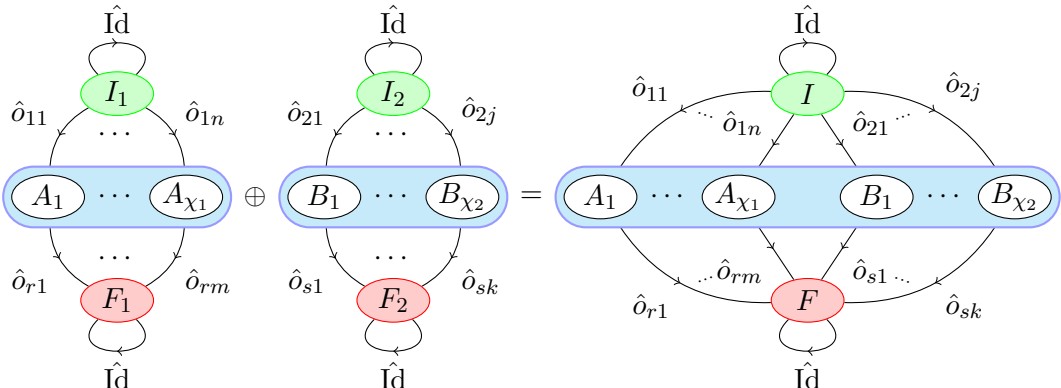

Figure 5: Realization of the operator sum $\hat{H}_1 + \hat{H}_2$ in terms of graph representations $\Lambda(\hat{H}_1 + \hat{H}_2) = \Lambda(\hat{H}_1) \oplus \Lambda(\hat{H}_2)$. Graph representations of operators $\hat{H}_{1,2}$ are illustrated by transitions from the initial state into the graph's bulk ($\hat{o}_{11} \ldots \hat{o}_{1n}$ and $\hat{o}_{21} \ldots \hat{o}_{2j}$) and from the graph's bulk to the final state ($\hat{o}_{r1} \ldots \hat{o}_{rm}$ and $\hat{o}_{s1} \ldots \hat{o}_{sk}$). Blue boxes denote the bulk of the graph representations $\Lambda(\hat{H}_{1,2})$ and $\Lambda(\hat{H}_1 + \hat{H}_2)$.

emphasize that each branch $\hat{H}_{\nu,r}$ generates exactly one type of local-operator string $\hat{o}_{\nu_1} \ldots \hat{o}_{\nu_r}$ and therefore the tensor representation of these strings factorizes on the local Hilbert spaces. Hence, if we can give a factorization of the local transformation $\hat{U}$ in terms of tensors acting on the local Hilbert spaces (e.g., $\hat{u}_{\nu_1} \ldots \hat{u}_{\nu_r}$), then we can represent the transformation directly by contracting the transformation tensors with the local operators over their physical indices

$$\hat{o}_{\nu_1} \ldots \hat{o}_{\nu_r} \longrightarrow \left[\hat{u}_{\nu_1} \hat{o}_{\nu_1}\right] \ldots \left[\hat{u}_{\nu_r} \hat{o}_{\nu_r}\right]. \tag{24}$$

Note that the transformations in eq. (13) forcing conservation of $U(1)$ quantum numbers for two-site gates are of exactly this kind and so is their generalization to arbitrary strings of local operators. Thus, conservation of $U(1)$ quantum numbers can be implemented via an initial transformation of the graph into its maximally branched representation and subsequent application of the shift tensors $\hat{R}_\Delta$ onto the local operator strings. For example, let us consider the transformation for a next-to-nearest-neighbor spin-flip term

$$\hat{S}^+_{(i)} \otimes \hat{\text{Id}}^{(i+1)} \otimes \hat{S}^-_{(i+2)} \quad \longrightarrow \quad \hat{S}^+_{(i)} \otimes \left[\hat{R}^{(i+1)}_{+1} \hat{\text{Id}}^{(i+1)}\right] \otimes \left[\hat{R}^{(i+2)}_{+1} \hat{S}^-_{(i+2)}\right] \tag{25}$$

with $\hat{S}^\pm$ being the usual angular-momentum ladder operators with lower site indices. The transformed graph representation is given in fig. 6a.

Conveniently, these rules can be extended to anticommuting operators by applying a Jordan-Wigner transformation.[4] For $U(1)$-invariant operators, the Jordan-Wigner transformation is also local, as there has to be a corresponding annihilation operator for every fermionic creation operator appearing in a string operator – and vice versa – because of quantum-number conservation. The Jordan-Wigner transformation can be implemented as a product of parity operators via

$$\hat{a}_{(i)} \rightarrow \hat{a}_{(i)} e^{i\pi \sum_{j<i} \hat{n}^{(j)}} = \prod_{j<i} \hat{\Pi}^{(j)} \hat{a}_{(i)}, \tag{26}$$

with $\hat{a}^{(\dagger)}_{(i)}$ annihilation (creation) operators for hard-core bosons at site $i$. Then we find that, for any $U(1)$-conserving product of fermionic creation and annihilation operators, the transformations act only within the operator strings. For instance, for a next-to-nearest-neighbor-hopping

---

[4]Note that this argument also holds in higher dimensions, because in MPS approaches a 1D path is used to sweep through the system, and the Jordan-Wigner transformation is also applicable in the presence of long-range interactions.

(a)                                                                  (b)

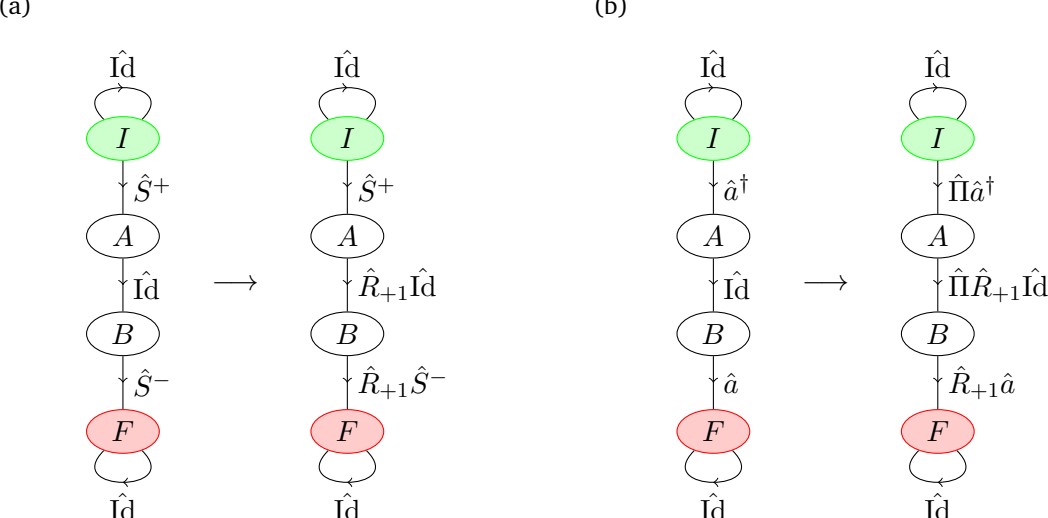

Figure 6: (a) Transformation of local operators $\hat{S}^{\pm}, \hat{\mathrm{Id}}$ to conserve $U(1)$ quantum numbers in the graph representation $\Lambda(\hat{H}_1)$ for a single-branch string operator $\hat{H}_1 = \sum_i \hat{S}^+_{(i)} \hat{S}^-_{(i+2)}$. (b) Transformation of local operators $\hat{a}, \hat{a}^{\dagger}, \hat{\mathrm{Id}}$ to conserve $U(1)$ quantum numbers in the graph representation $\Lambda(\hat{H}_2)$ for a single-branch string operator, which describes fermionic next-to-nearest-neighbor hopping $\hat{H}_2 = \sum_i \hat{f}^{\dagger}_{(i)} \hat{f}_{(i+2)}$ (with $\left\{ \hat{f}_{(i)}, \hat{f}^{\dagger}_{(j)} \right\} = \delta_{ij}$).

term we find

$$\hat{a}^{\dagger}_{(i)} \otimes \hat{\mathrm{Id}}^{(i+1)} \otimes \hat{a}_{(i+2)} \quad \longrightarrow \quad \left[ \hat{\Pi}^{(i)} \hat{a}^{\dagger}_{(i)} \right] \otimes \left[ \hat{\Pi}^{(i+1)} \hat{R}^{(i+1)}_{+1} \hat{\mathrm{Id}}^{(i+1)} \right] \otimes \left[ \hat{R}^{(i+2)}_{+1} \hat{a}_{(i+2)} \right]. \qquad (27)$$

Again, these transformations have a simple graph representation, see fig. 6b.

To sum up, we have derived a construction scheme for MPO representations of generic $U(1)$-invariant operators that takes a FSM as input. Specifying the phase factor $e^{i\phi}$ for the commutation relations of the local operators (e.g., $\phi = 2\pi$ for bosons, $\pi$ for fermions), the scheme automatizes the construction of the MPO site tensors so that we can identify the graph representation of the FSM with the MPO. This permits us to take advantage of the graph representation to improve operator arithmetics, which will be discussed in the following section.

## 4 Graph arithmetics and MPO representation of the variance of operators

Having derived a construction scheme that permits us to automatize the generation of MPO representations for $U(1)$-invariant operators, we now make use of the established connection between graphs and operators by replacing operator arithmetics with graph manipulations. We then demonstrate the power of this approach by employing the graph algebra to derive various expressions for the variance of a Hamiltonian $\hat{H}$. As we will show, this not only allows for more efficient calculations, but also addresses the problem of catastrophic cancellation that comes up in a naïve evaluation of $var(\hat{H}) = \left\langle \hat{H}^2 \right\rangle - \left\langle \hat{H} \right\rangle^2$ due to the need to subtract large numbers, which scale roughly as $\mathcal{O}(L^2)$ in the system size $L$ [19].

Let us consider the product of two global operators $\hat{H}_1, \hat{H}_2$ in terms of their maximally

branched representations

$$\hat{H}_1 \cdot \hat{H}_2 = \left( \sum_{\nu^1, r^1} \hat{H}_{\nu^1, r^1} \right) \cdot \left( \sum_{\nu^2, r^2} \hat{H}_{\nu^2, r^2} \right) \tag{28}$$

and in particular a single summand that is the product of two lattice-ordered string operators[5]

$$\hat{H}_{\nu^1, r^1} \cdot \hat{H}_{\nu^2, r^2} = \left( \sum_i \hat{h}^{(i)}_{\nu^1, r^1} \right) \cdot \left( \sum_j \hat{h}^{(j)}_{\nu^2, r^2} \right) . \tag{29}$$

Note that we have introduced superscripts $\nu^{1,2}, r^{1,2}$ to distinguish the index sets of the global operator.[6] A representation of this product in terms of a FSM and therefore its graph representation requires a reformulation in terms of lattice-ordered string operators. Although the product of two lattice-ordered operators $\hat{H}_{r^1, \nu^1} \cdot \hat{H}_{\nu^2, r^2}$ is no longer lattice-ordered, a careful inspection of the terms violating the lattice order reveals how to build a graph representation generating the product $\hat{H}_{r^1, \nu^1} \cdot \hat{H}_{r^2, \nu^2}$. It turns out to be useful to define a non-commutative $\wedge$ product, which maps two single-branched graphs to a single-branched graph via

$$\Lambda(\hat{H}_{\nu^1, r^1}) \wedge \Lambda(\hat{H}_{\nu^2, r^2}) = \Lambda \left( \sum_{i + r^1 < j} \hat{h}^{(i)}_{\nu^1, r^1} \hat{h}^{(j)}_{\nu^2, r^2} \right) . \tag{30}$$

A graph realization of $\wedge$ is obtained by identifying the final state of $\Lambda(\hat{H}_{\nu^1, r^1})$ with the initial state of $\Lambda(\hat{H}_{\nu^2, r^2})$. For instance, see fig. 7a for an exemplary evaluation of $\Lambda(\sum_i \hat{o}^{(i)}_{11} \hat{o}^{(i+1)}_{12}) \wedge \Lambda(\sum_j \hat{o}^{(j)}_{21} \hat{o}^{(j+1)}_{22})$.

As carried out in appendix A, an algorithm can be constructed that yields the following graph representation

$$\Lambda(\hat{H}_{r^1, \nu^1} \cdot \hat{H}_{r^2, \nu^2}) = \left[ \Lambda(\hat{H}_{r^1 \nu^1}) \wedge_S \Lambda(\hat{H}_{r^2, \nu^2}) \right] \oplus \bigoplus_{\Delta = 0}^{r^1 + r^2} \hat{\gamma}_\Delta \equiv \Lambda(\hat{H}_{r^1, \nu^1}) \otimes \Lambda(\hat{H}_{r^2, \nu^2}) , \tag{31}$$

with the symmetrized wedge product

$$\Lambda(\hat{H}_{r^1 \nu^1}) \wedge_S \Lambda(\hat{H}_{r^2, \nu^2}) \equiv \left[ \Lambda(\hat{H}_{r^1, \nu^1}) \wedge \Lambda(\hat{H}_{r^2, \nu^2}) \right] \oplus \left[ \Lambda \left( \text{sgn}(\hat{h}_{\nu^1, r^1}, \hat{h}_{\nu^2, r^2}) \hat{H}_{r^2, \nu^2} \right) \wedge \Lambda(\hat{H}_{r^1, \nu^1}) \right] , \tag{32}$$

and $\text{sgn}(\hat{h}_{\nu^1, r^1}, \hat{h}_{\nu^2, r^2})$ the sign of the commutation relation between local operators $\hat{h}_{\nu^1, r^1}$ and $\hat{h}_{\nu^2, r^2}$ acting on different sites. $\bigoplus_{\Delta = 0}^{r^1 + r^2} \hat{\gamma}_\Delta$ includes all overlapping branches and is explained in detail in appendix A. Employing linearity of the graph representation for addition, the product of two general operators can then be formulated via

$$\begin{aligned}
\Lambda(\hat{H}_1 \cdot \hat{H}_2) &= \Lambda \left( \sum_{r^1, \nu^1} \sum_{r^2, \nu^2} \hat{H}_{r^1, \nu^1} \cdot \hat{H}_{r^2, \nu^2} \right) \\
&= \bigoplus_{\nu^1, r^1} \bigoplus_{\nu^2, r^2} \left[ \Lambda(\hat{H}_{r^1, \nu^1}) \otimes \Lambda(\hat{H}_{r^2, \nu^2}) \right] .
\end{aligned} \tag{33}$$

Despite the compact form, we emphasize that eq. (31) and eq. (33) describe a graph representation that is maximally expanded, so that there is no branching below the initial node, and the bond dimension of the generated MPO is very large. However, the size of the graph can be reduced very efficiently by shrinking it into its most compact form. The idea behind

---

[5]We use the notation introduced in eq. (19).

[6]Expanding the indices one would have $\hat{H}_i = \sum_{\nu^i_1 \dots \nu^i_{n_i}, r^i} \hat{H}_{\nu^i_1 \dots \nu^i_{n_i} r^i}$

(a)

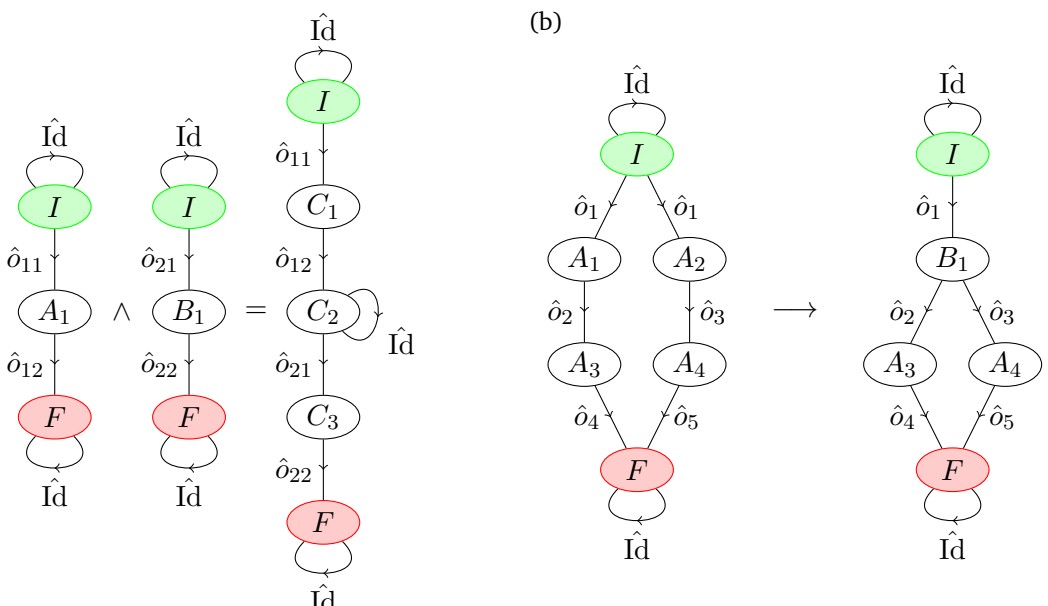

Figure 7: (a) Graph representation of $\Lambda(\hat{H}_{\nu^1, r^1}) \wedge \Lambda(\hat{H}_{\nu^2, r^2})$ for 2-point 2-ranged interacting operator strings. (b) Merging of edges in $\Lambda(\hat{A}) = \Lambda(\hat{o}_1 \hat{o}_2 \hat{o}_4) \oplus \Lambda(\hat{o}_1 \hat{o}_3 \hat{o}_5)$ to reduce the total number of required states.

the shrinking is best demonstrated with a concrete example. Consider an expanded graph generated from merging two branches

$$\Lambda(\hat{A}) = \Lambda(\hat{o}_1 \hat{o}_2 \hat{o}_4) \oplus \Lambda(\hat{o}_1 \hat{o}_3 \hat{o}_5). \tag{34}$$

The number of nodes can be reduced by fusing edges sharing one node and carrying the same transitions. For example, in fig. 7b these are the edges connecting the states $I \rightarrow A_1$ and $I \rightarrow A_2$. In the same way, entire branches that are completely equal with respect to their transitions can be fused together by employing the linearity of $\oplus$

$$\Lambda(\hat{A}) = \Lambda(\hat{o}_1 \hat{o}_2) \oplus \Lambda(\hat{o}_1 \hat{o}_2) = \Lambda(2\hat{o}_1 \hat{o}_2). \tag{35}$$

But we can go even further by defining which local operator pairs acting on the same lattice site vanish identically. For example, this is the case for $\hat{S}^+ \hat{S}^+ \equiv 0$ for $S = 1/2$ models. Generated branches that contain this type of transitions can be discarded completely. For more complex graphs, these shrinking procedures can be applied iteratively, so that only the reduced graph is stored and used in the actual calculations.

An illustrative example demonstrating the advantages of the approach above is to construct two different expressions for the variance of the Hamiltonian $\hat{H}$ of a system, where the goal is to avoid catastrophic cancellation as best as possible while keeping calculations as cheap as possible. For instance, the variance can be used as a control parameter in numerical simulations to test whether a state $|\psi\rangle$ is close to an eigenstate of $\hat{H}$. A naïve evaluation is obtained by directly calculating the expectation values in

$$var(\hat{H}) = \langle\psi|\hat{H}^2|\psi\rangle - \langle\psi|\hat{H}|\psi\rangle^2 \geq 0, \tag{36}$$

which eventually vanishes, if $|\psi\rangle$ is an exact eigenstate. However, as the expectation value of $\hat{H}^2$ scales as $L^2$, explicit evaluation of eq. (36) has major drawbacks when it comes to numerical

calculations with finite-precision arithmetics such as catastrophic cancellation. Every (exact) number $z$ is numerically represented up to a certain precision [20], which is usually measured in orders $p$ of magnitudes so that we can mimic limited numerical precision by replacing $z \to z(1 + \epsilon \times 10^{-p})$ with a random variable $\epsilon \in (-1, 1)$. Let $z_{1,2}$ be numbers represented with the same numerical precision $p$ and $0 < z_1 - z_2 = 10^{-\delta}$ their exact difference. In finite-precision arithmetics, we then obtain

$$
\begin{aligned}
z_1 - z_2 &\stackrel{\sim}{=} z_1(1 + \epsilon_1 \times 10^{-p}) - z_2(1 + \epsilon_2 \times 10^{-p}) \\
&= 10^{-\delta} + \epsilon_1 \times 10^{-(p+\delta)} + z_2(\epsilon_2 - \epsilon_1) \times 10^{-p}.
\end{aligned}
\tag{37}
$$

For $\delta > 0$, the second term cannot be represented due to finite precision. In our case, the values for $z_{1,2}$ are obtained from expectation values of operators acting on the whole system. Hence, if we estimate them by their leading-order contribution $z_{1,2} \sim L^q$ with magnitude $q$, with $\gamma \equiv \log_{10}(L)$, and $\epsilon \equiv \epsilon_2 - \epsilon_1$, we obtain

$$
z_1 - z_2 \stackrel{\sim}{=} 10^{-\delta} + \epsilon \times 10^{-(p-\gamma \cdot q)}
\tag{38}
$$

where $\epsilon$ is a random variable of order $\pm 1$. It follows that we need $\delta < p - \gamma \cdot q$ in order to have a reasonable numerical outcome. In case of the variance, i.e., $q = 2$, with double-precision arithmetics, $p = p_{\text{num}} = 16$, the naïve evaluation, $\delta < 16 - 2\gamma$, yields an upper bound of a maximally possible precision of $10^{-12}$ for a lattice with $L = 100$ sites. However, the numerical precision in general is not only bound by the exact numerical precision $p_{\text{num}}$, but it is also subject to round-off errors of preceding calculations. Thus, in actual calculations, we have an effective precision that depends on simulation parameters $p(L, \chi, ...) \leq p_{\text{num}}$. Returning to eq. (36), we realize that the calculated variance is not only bound, but may even become negative (as $\epsilon$ in eq. (38) can be negative).

The problem can be addressed by a minimization of $\gamma \cdot q$, for instance by constructing an MPO representation for the operator $\hat{V} = (\hat{H} - \langle\psi| \hat{H} |\psi\rangle)^2$, (e.g., by distributing the expectation value $\langle\psi| \hat{H} |\psi\rangle$ over the lattice sites). Unfortunately, this comes at the cost of having constructed an operator that is dependent on its expectation value. Hence, its MPO representation has to be rebuilt for every new state $|\psi\rangle$, which requires an efficient way of obtaining MPO representations of powers of operators while keeping the numerical effort low.

To analyze both problems, we investigate two MPO representations $\Lambda_1(var(\hat{H}))$ (see fig. 8a for the corresponding graph) and $\Lambda_2(var(\hat{H}))$, which are obtained from the graph representations

$$
\begin{aligned}
\Lambda_1(var(\hat{H})) &= \Lambda(\hat{H}^2) \oplus \Lambda(-E^2 \hat{\text{Id}}) \\
&= \Lambda(\hat{H}^2) \oplus \left[ \Lambda(-\epsilon_0 \hat{\text{Id}}) \wedge \Lambda(\epsilon_1 \hat{\text{Id}}) \right], \qquad \text{with} \quad \epsilon_i = \langle \hat{H} \rangle \frac{\sqrt{2}}{L - i}
\end{aligned}
\tag{39}
$$

$$
\begin{aligned}
\Lambda_2(var(\hat{H})) &= \Lambda(\hat{H} - \langle\hat{H}\rangle) \otimes \Lambda(\hat{H} - \langle\hat{H}\rangle) \\
&= \left[ \Lambda(\hat{H}) \oplus \Lambda(\epsilon \hat{\text{Id}}) \right]^{\otimes 2}, \qquad \text{with} \quad \epsilon = -\frac{\langle\hat{H}\rangle}{L}.
\end{aligned}
\tag{40}
$$

Let us briefly discuss the properties of the two graph representations and their generated MPO. We start with the numerical implementation and its costs. Both representations are obtained by loading pre-computed graph products at runtime. Then, we explicitly calculate $\langle\hat{H}\rangle$ with numerical costs scaling as $\mathcal{O}(LdD^2D_W^2)$ with $D$ and $D_W$ being the maximal matrix dimensions of the MPS and MPO site tensors, respectively. Construction of the MPO representation for $\Lambda_{1,2}(var(\hat{H}))$ then only requires allocation of the site tensors with costs scaling as $\mathcal{O}(L[dD_{W_{1,2}}]^2)$. Here, $D_{W_{1,2}}$ denotes the maximal bond dimensions of the MPO site-tensor matrices generated from the graphs $\Lambda_{1,2}(var(\hat{H}))$. Subsequently, the MPO tensors of the graphs are compressed by using an SVD with numerical costs scaling as $\mathcal{O}(L \cdot [d^2 \cdot D_{W_{1,2}}]^3)$.

(a)

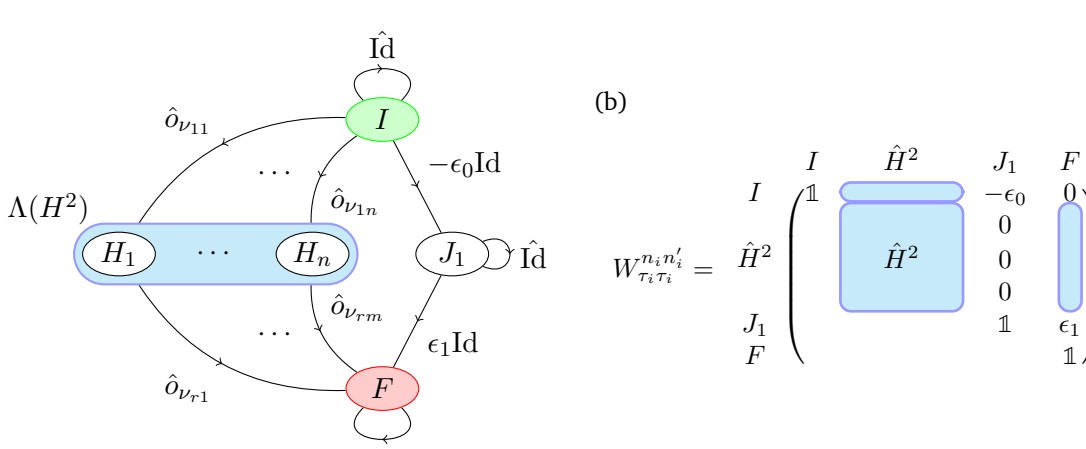

Figure 8: (a) Graph representation $\Lambda_1(var(\hat{H}))$, compare eq. (39). The representation of $\hat{H}^2$ and of $\langle\hat{H}\rangle^2$ are considered independently: For the former, we depict transitions from the initial state to the subgraph $\Lambda(\hat{H}^2)$ and then to the final state indicated by the operators $\hat{o}_i$, ($\Lambda(\hat{H}^2)$ represents the graph of $\hat{H}^2$). For the latter, we depict transitions from the initial state to the state $J_1$ of the FSM, and then to the final state via identity operators. Those identities carry the weights $-\epsilon_0$ and $\epsilon_1$, respectively, which after performing the multiplication yields $-\langle\hat{H}\rangle^2$, see eq. (39). (b) MPO site tensor obtained from the graph representation $\Lambda_1(var(\hat{H}))$. Matrix entries with blue background denote the entries obtained from the site-tensor representation of $\Lambda(\hat{H}^2)$.

To sum up, the numerical costs for obtaining the MPO representation of the graphs $\Lambda_{1,2}(var(\hat{H}))$ are to leading order governed by the expenses when calculating the expectation value $\langle\hat{H}\rangle$, as long as the MPS bond dimensions are the cost-determining factor. This demonstrates a major advantage of mapping the MPO arithmetics onto the graph representations: As the expectation value $\langle\hat{H}\rangle$, in general, is already available, the additional numerical costs for constructing the MPO representation are not significant.

Next, we take a look at the numerical stability. The graph $\Lambda_2(var(\hat{H}))$ generates the MPO representation of the operator $\langle\hat{H}-\langle\hat{H}\rangle\rangle$ multiplied by itself. Thus, the graph of $\hat{H}$ is expanded so that the expectation value $\langle\hat{H}\rangle$ is equally distributed over all lattice sites with an on-site value of $\langle\hat{H}\rangle/L$. As operator arithmetics are represented exactly by means of the constructed graph $\Lambda_2(var(\hat{H}))$, the only relevant source of catastrophic cancellation is the evaluation of $\langle\hat{H}-\langle\hat{H}\rangle\rangle$ along the lattice that compares terms of order $\mathcal{O}(L)$, hence $q=1$. Thus, we expect the variance to be bound from below by $10^{16-\gamma}$. Yet, the graph $\Lambda_1(var(\hat{H}))$ in general also suffers from catastrophic cancellation with $q=2$, as in the naïve evaluation of the variance. This is best seen by evaluating the structure of the generated matrix representation for non-vanishing tensor blocks $W^{n_i n_i'}_{\tau_{i-1}\tau_i}$ (see fig. 8b). As the latter contains the complete matrix representation of $\hat{H}^2$, we end up at the final site by comparing numbers of order $\mathcal{O}(L^2)$ when performing the tensor contractions to evaluate the variance. Therefore, $q=2$ yields the variance to be bound from below by $10^{16-2\gamma}$.

In addition, graph arithmetics are exact, whereas MPO arithmetics need a vast amount of numerical operations completed before expectation values can be calculated. Therefore, using MPO arithmetics is much more prone to collecting round-off errors, which, in drastic cases, reduces the numerical precision $p$ by orders of magnitudes [5].

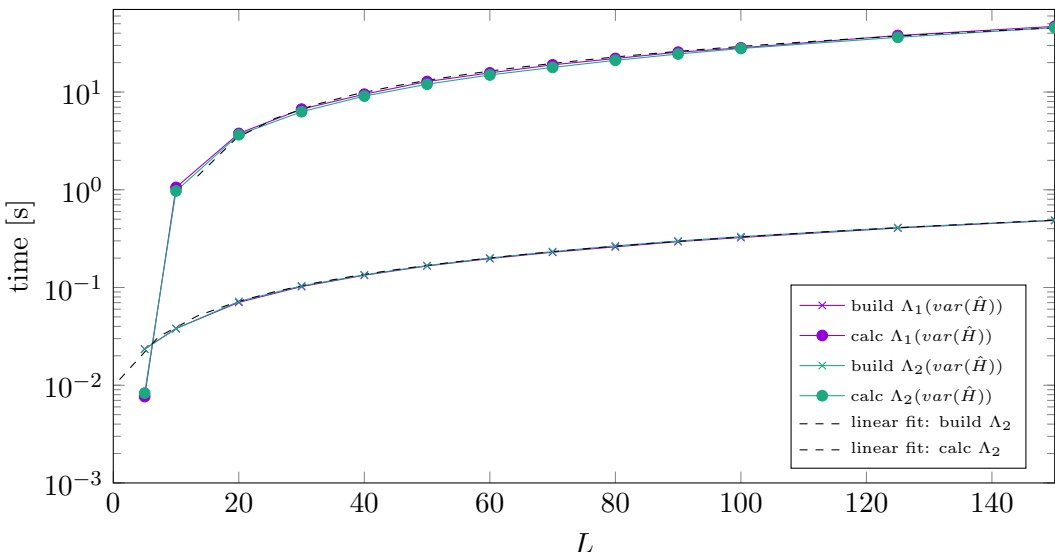

Figure 9: Computational time to construct and calculate the variance via graph representations $\Lambda_{1,2}(var(\hat{H}_{\text{Heisenberg}}))$ for a $S = 1$ Heisenberg chain in a random state with matrix dimension $D = 100$ in every non-vanishing MPS site-tensor block.

# 5 Numerical behavior of the variance for antiferromagnetic $S = 1$ Heisenberg chains

In this section, we test the behavior of the variance obtained from site-tensor representations of $\Lambda_{1,2}(var(\hat{H}_{\text{Heisenberg}}))$ for $S = 1$ antiferromagnetic Heisenberg chains [21, 22]

$$\hat{H}_{\text{Heisenberg}} = \sum_i \hat{\mathbf{S}}_i \cdot \hat{\mathbf{S}}_{i+1}\,, \tag{41}$$

with $\hat{\mathbf{S}}_i$ the vector of $S = 1$ spin operators on lattice site $i$. For this purpose, we exploit the total magnetization of a system with $L$ lattice sites as the conserved $U(1)$ quantity. For the ground state search, we sweeps through the system (using open boundary conditions) and optimize the site tensors via a standard Lanczos algorithm, see [23–25]. All MPS contractions are formulated in two-site representation [5].

The largest observed bond dimensions $D_{W_{1,2}}$ of the site-tensor matrices $W^{n_i n_i'}_{\tau_{i-1} \tau_i}$ for the MPO representations of $\Lambda_{1,2}(var(\hat{H}_{\text{Heisenberg}}))$ are $D_{W_1,max} = 12$ and $D_{W_2,max} = 10$, respectively. Note that these bond dimensions are generally smaller than MPS bond dimensions used during the simulation. We conclude from $D_{W_2,max} < D_{W_1,max}$ that the distribution of the constant energy term $\langle \hat{H}_{\text{Heisenberg}} \rangle$ over the lattice sites in $\Lambda_2(var(\hat{H}_{\text{Heisenberg}}))$ ensures a more efficient compression of the resulting MPO representation. Consistently, for common MPS bond dimensions, we find that $\Lambda_2(var(\hat{H}_{\text{Heisenberg}}))$ is evaluated faster than $\Lambda_1(var(\hat{H}_{\text{Heisenberg}}))$. In addition, fig. 9 displays the build time of the MPO representation from the graphs $\Lambda_{1,2}(var(\hat{H}_{\text{Heisenberg}}))$ for various system sizes. We find a perfect linear scaling $\sim \mathcal{O}(L)$ for the dependency of the build time, which moreover has no impact on the overall computation time.

We have performed simulations, in which we varied either the total bond dimension per MPS site tensor $\chi_{\max}$ or the discarded weight $w$ while keeping the respective other fixed.[7]

---

[7] We define the total bond dimension $\chi_{\max}$ as the number of singular values kept per site and the discarded weight as the sum over all squares of neglected singular values $w = \sum_{k=\chi_{\max}+1}^{D} S_k^2$.

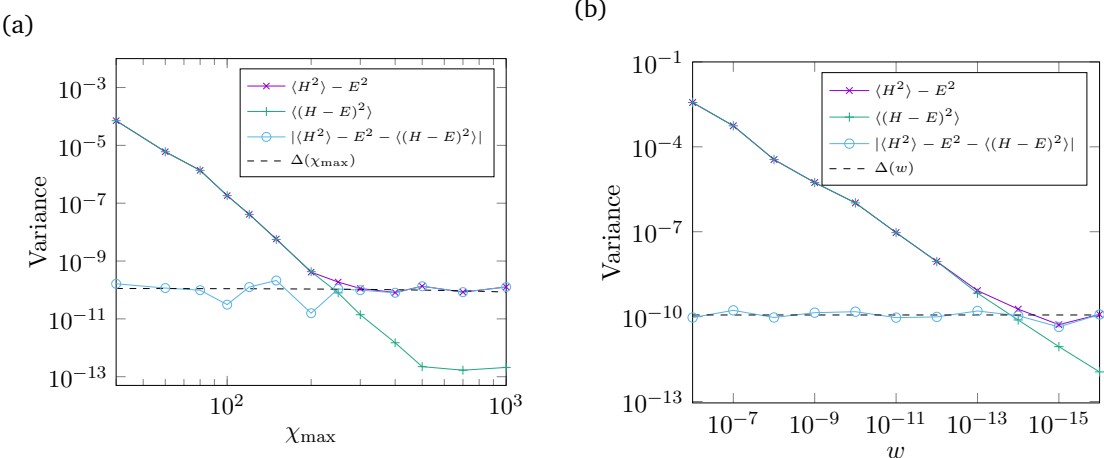

Figure 10: (a) Variance as function of the maximum total bond dimension $\chi_{\max}$ for a $S = 1$ Heisenberg chain with 100 sites and discarded weight $w = 0$ in the ground state. $\Delta(\chi_{\max})$ is obtained from a linear fit of the difference $|\langle H^2 \rangle - E^2 - \langle (H-E)^2 \rangle|$, which yields $\Delta(\chi_{\max}) = -2.88607 \times 10^{-14} \cdot \chi_{\max} + 1.1342 \times 10^{-10}$. (b) Variance as function of the discarded weight $w$ for a $S = 1$ Heisenberg chain with 100 sites and $\chi_{\max} = 500$ in the ground state. $\Delta(\chi_{\max})$ is obtained from a linear fit of the difference $|\langle H^2 \rangle - E^2 - \langle (H-E)^2 \rangle|$, which yields $\Delta(w) = 1.14249 \times 10^{-10}$.

Figures 10a and 10b show results for various values of $\chi_{\max}$ and $w$. An important criterion for consistency of the calculations is the independence of the threshold at which catastrophic cancellation sets in. We find that varying both parameters $\chi_{\max}$ and $w$ yields a constant value of

$$\Delta \equiv |\langle H^2 \rangle - E^2 - \langle (H-E)^2 \rangle| \cong 10^{-10}, \tag{42}$$

which corresponds to the value at which the graph representation $\Lambda_1(var(\hat{H}_{\text{Heisenberg}}))$ saturates. Employing eq. (38), these results suggest for the actual numerical precision an ansatz of the form $p = p_{\text{num}} - p_r(L)$ with a correction $p_r(L)$, which to first order depends only on the lattice size $L$. Repeating the calculations for $S = 1$ Heisenberg chains with various system sizes, we can thus extract $p_r$ from eq. (38) by estimating $\delta = \log_{10}(var(\hat{H}_{\text{Heisenberg}}))$ from the saturated value for the variances. For the two graph representations we consider the estimator for the actual numerical precision

$$p_{1,2}(L) = -\delta_{1,2}(L) - \gamma(L)q_{1,2}. \tag{43}$$

For both graphs, we perform a linear fit obtaining

$$p_1 = (15.6 \pm 0.4) - (1.1 \pm 0.2) \cdot \gamma(L) \tag{44}$$

$$p_2 = (16 \pm 1) - (1.2 \pm 0.7) \cdot \gamma(L), \tag{45}$$

which is shown in Fig. 11. We emphasize that the observed behavior is perfectly consistent with the ansatz above for constant double-precision arithmetics $p_{\text{num}} = 16$ and residual numeric precision $p_r(L) \approx \gamma(L) = \log_{10}(L)$. We hence find that, aside from catastrophic cancellation, the dominating contribution to the loss of numerical precision is proportional to the lattice size, which can be associated with inevitable rounding errors generating an error $\mathcal{O}(10^{-p_{\text{num}}})$ per lattice site.

To complete this section, we now turn to the method discussed by Hubig et al. in [14], which suggests a complementary access for introducing operator arithmetics for MPOs. In

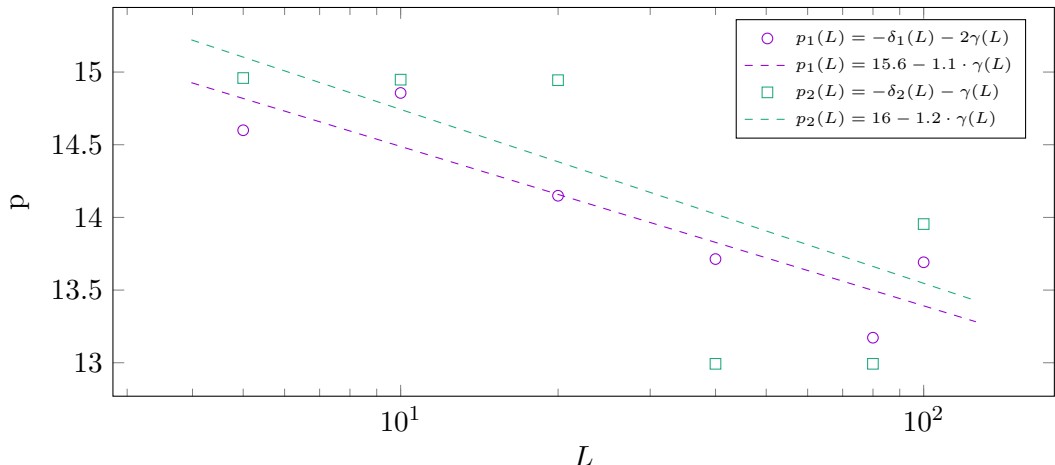

Figure 11: Numerical precision calculated from eq. (38) for graph representations $\Lambda_{1,2}(var(\hat{H}_{\text{Heisenberg}}))$ evaluated for various lattice sizes. The dashed lines are linear fits that illustrate the average dependence on the lattice size of the contribution $p_r(L)$ of the effective numerical precision in the tensor-network contractions, $p(L) = p_{\text{num}} - p_r(L)$.

short, Hubig et al. present a scheme to build MPO representations numerically via in-code evaluation of direct sums and Kronecker products of local operator strings acting on individual lattice sites. To account for the growth in the MPO bond dimension, they discuss various numerical compression schemes and benchmark the obtained MPOs for a 100-site $S = 1$ Heisenberg chain. This scheme is somewhat more straightforward, as it only involves elementary matrix operations.

However, the FSM approach has additional advantages when it comes to numerical calculations. In particular, when comparing our results for the variance with those obtained by Hubig et al. for the same numerical simulation parameters,[8] we find that the maximally obtained precision is two orders of magnitudes larger when using the MPOs obtained from graph arithmetics. The reason lies in the above observation that the graph representations only require a minimal number of floating-point operations. In fact, the MPO representation for $var(\hat{H})$ constructed from numerical MPO arithmetics as suggested by Hubig et al. already requires at least $L$ Kronecker products and sums to be evaluated for the advantageous MPO representation of the variance. In contrast, the FSM is generated from abstract graph operations and is therefore an *exact* representation.

Thus, before the actual variance calculation (namely the contraction of the tensor network representation of $\langle\psi|var(\hat{H})|\psi\rangle$) can begin, the MPO representation obtained from numerical MPO arithmetics already picked up numerical round-off errors of magnitude $L = 10^{\gamma}$. Therefore, in the example of the 100-site $S = 1$ Heisenberg chain, the decrease in precision can be estimated to be $p_{\text{loss}} \sim \gamma = 2$ magnitudes, which is exactly what we found in our calculations.

Another nice side effect is that performing graph compression as described in section 4 already covers the *deparallelisation* compression method which was first introduced in [26].

Finally, note that implementing local transformations on an abstract level as discussed in section 3 maps the whole complexity of index shifting, as required by quantum-number conservation or in an implementation of fermionic anticommutation rules, from the code to an input level. In other words, there is no need for the programmer to hard-code these features when MPOs are generated from FSMs. Instead, they can be incorporated by designing the

---

[8]Note that Hubig et al. employ conservation of non-abelian quantum numbers. Therefore, the MPS bond dimensions used there do not directly compare to the ones used here. However, this does not affect the maximally achievable resolution of $\langle var(\hat{H})\rangle$ due to finite precision arithmetics.

corresponding graphs, which in turn enormously increases the flexibility of the code.

## 6 Conclusions

We have formulated an optimized algorithmic construction scheme for efficient MPO representations of $U(1)$-invariant operators generated by FSMs. This scheme allows implementations to automatize the application of local transformations of operators in MPO representations, e.g., while exploiting $U(1)$ symmetries by propagating local changes in quantum numbers or by tackling the fermionic sign via a Jordan-Wigner transformation. As a consequence, graph representations for FSMs can be interpreted directly as representations of operators. Based on this, operator arithmetics are then mapped to transformations of the underlying graphs to generate exact MPO representations of operator sums and products. This permits us to exactly calculate operator arithmetics, which can be stored and quickly loaded in the course of simulations. We demonstrated the effectiveness of our approach by considering two graph representations $\Lambda_{1,2}(var(\hat{H}))$ of the variance of a system's Hamiltonian $\hat{H}$. Investigating their numerical properties in a ground-state calculation for a $S = 1$ Heisenberg chain with $L = 100$ lattice sites, both representations behave numerically consistent and stable with a resolution for the variances $var_{1,2}(\hat{H}_{\text{Heisenberg}})$ of the graph representations of $var_1(\hat{H}_{\text{Heisenberg}}) \overset{\sim}{=} \mathcal{O}(10^{-10})$ and $var_2(\hat{H}_{\text{Heisenberg}}) \overset{\sim}{=} \mathcal{O}(10^{-12})$, respectively. Investigating the dependence of the numerical breakdown on the lattice size shows that the graph representations achieve a numerical precision of $p \overset{\sim}{=} \mathcal{O}(p_{\text{num}} - \log_{10}(L))$ in the calculations. We conclude that this high numerical precision is due to the exact graph representation of the operator arithmetics and comes without any significant additional computational costs during runtime.

Finally, we note that wrapping operator arithmetics into re-usable graph representations helps to obtain efficient and exact MPO representations, which are useful for various applications. For instance, consider variational problems of the form

$$\min_{|\psi\rangle} \langle \psi | (\hat{H} - E)^2 | \psi \rangle - \lambda \langle \psi | \psi \rangle \tag{46}$$

to find highly excited eigenstates and eigenvalues. Solving this minimization problem benefits significantly from the increased numerical precision and gains in efficiency of the calculation of the expectation value of $(\hat{H} - E)^2$.

Further applications of the introduced approach cover the efficient representation of long-ranged swap gates, and the concepts can be generalized to higher-dimensional tensor networks (e.g. PEPS) [27]. In this case, the transition amplitudes get additional degrees of freedom ('color'), corresponding to either transversal or longitudinal auxiliary indices. Similar benefits are to be expected as in 1D MPS, allowing for more precise and more flexible implementations of tensor network methods in higher dimensions.

## Acknowledgements

We thank M. Marahrens and B. Lenz for helpful discussions, and I. Köhler for carefully proofreading the manuscript. We gratefully acknowledge financial support by the Deutsche Forschungsgemeinschaft (DFG) through Research Unit FOR 1807 (project P7) and SFB/CRC 1073 (project B03).

## A  Construction of a graph representation for lattice-ordered string operator products

Consider two lattice-ordered string operators $\hat{H}_{\nu^{1,2},r^{1,2}}$ as introduced in the main text (eq. (19)) with single-branch graph representations $\Lambda(\hat{H}_{\nu^1,r^1})$ and $\Lambda(\hat{H}_{\nu^2,r^2})$, which are parts of maximally branched operators $\hat{H}_{1,2}$. In the following, we present an algorithm to construct the graph representation of the operator product $\hat{\tilde{H}} = \Lambda(\hat{H}_{\nu^1,r^1}) \cdot \Lambda(\hat{H}_{\nu^2,r^2})$ in terms of generating a new graph. This procedure can then be applied to all branches to construct a new graph for $\hat{H} = \hat{H}_1 \cdot \hat{H}_2$.

Let $\mathcal{T}(\mathcal{K}, b = 1, n)$ be the set of all single-branch graphs representing lattice-ordered $n$-point operators. Then, with a proper $\beta \geq b$, we look for a realization of the non-commutative map

$$\otimes : \mathcal{T}(\mathcal{K}, 1, n) \times \mathcal{T}(\mathcal{K}', 1, m) \longrightarrow \mathcal{T}(\mathcal{K} \times \mathcal{K}', 2 + \beta, n + m)$$
$$\Lambda(\hat{H}_{\nu^1,r^1}) \otimes \Lambda(\hat{H}_{\nu^2,r^2}) \longmapsto \Lambda(\hat{H}_{\nu^1,r^1} \cdot \hat{H}_{\nu^2,r^2}) , \tag{47}$$

with $\mathcal{K} \times \mathcal{K}'$ denoting the symmetrized on-site tensor-product set of $\mathcal{K}$ and $\mathcal{K}'$. For this purpose, we apply the definition of $\oplus$ in eq. (22) and search for a graph representation of $\Lambda(\hat{H}_{\nu^1,r^1} \cdot \hat{H}_{\nu^2,r^2})$ by ordering the appearing types of terms in the resulting sum of the operator product according to the lattice treated. We construct single-branch graph representations for all different types of generated lattice-ordered operator strings, which we denote by $\hat{\gamma}$. Then, a graph representation is obtained by summing up all these strings

$$\Lambda(H_{\nu^1,r^1} \cdot \hat{H}_{\nu^2,r^2}) = \bigoplus_{\hat{\gamma}} \Lambda(\hat{\gamma}). \tag{48}$$

From now on, we focus on the special case of 2-point operators, i.e., operators of the form

$$\hat{H}_{\nu^n,r^n} = \sum_{i_n} \hat{h}^{(i_n)}_{\nu_1^n \nu_2^n, r^n} = \sum_{i_n} f^{r^n}_{\nu_1^n \nu_2^n} \hat{o}^{(i_n)}_{\nu_1^n} \hat{o}^{(i_n+r^n)}_{\nu_2^n}. \tag{49}$$

Nevertheless, the generalization to arbitrary $n$-point string operators is straightforward: Simply replace identities with additional local operators. Decomposing the operator product, we find

$$\hat{H}_{r^1,\nu^1} \cdot \hat{H}_{r^2,\nu^2} = \sum_{i_1,i_2} \hat{h}^{(i_1)}_{\nu_1^1 \nu_2^1, r^1} \hat{h}^{(i_2)}_{\nu_1^2 \nu_2^2, r^2}$$
$$= \underbrace{\sum_{i_1} \sum_{i_2 > i_1 + r^1} \hat{h}^{(i_1)}_{\nu_1^1 \nu_2^1, r^1} \hat{h}^{r^2}_{\nu_1^2 \nu_2^2}(i_2)}_{\hat{H}_A =} + \underbrace{\sum_{i_1} \sum_{i_2 < i_1 - r^2} \hat{h}^{(i_1)}_{\nu_1^1 \nu_2^1, r^1} \hat{h}^{(i_2)}_{\nu_1^2 \nu_2^2, r^2}}_{\hat{H}_C =}$$
$$+ \underbrace{\sum_{i_1} \sum_{i_2 = i_1 - r^2}^{i_1 + r^1} \hat{h}^{(i_1)}_{\nu_1^1 \nu_2^1, r^1} \hat{h}^{(i_2)}_{\nu_1^2 \nu_2^2, r^2}}_{\hat{h}_B(i_1,i_2) \equiv \text{Overlaps}} . \tag{50}$$

Next, the commutation relation of local operators acting on different sites $i_1 \neq i_2$ fulfills

$$\left[\hat{o}^{(i_1)}_{\nu_1}, \hat{o}^{(i_2)}_{\nu_2}\right]_{\epsilon_{\nu_1 \nu_2}} = \hat{o}^{(i_1)}_{\nu_1} \hat{o}^{(i_2)}_{\nu_2} - \epsilon_{\nu_1 \nu_2} \hat{o}^{(i_2)}_{\nu_2} \hat{o}^{(i_1)}_{\nu_1} = 0, \tag{51}$$

with $\epsilon_{\nu_1 \nu_2}$ the sign according to the commutation relation. Then, we can decompose the product into lattice-ordered sums by commuting local operators acting on strictly unequal sites.

(a)  (b)  (c) $\Delta = 0$  (d) $\Delta = 1$

(e) $\Delta = r^1 + 1$ $(r^1 < r^2)$  (f) $\Delta = r^1 + r^2$

(g)

Figure 12: (a) Tree diagram representation of $\hat{H}_A$. (b) Tree diagram representation of $\hat{H}_C$. (c-f) The figures depict stages of the algorithm for determining all summands $\hat{h}_B(i_1, i_2)$ by shifting the operator string $\hat{h}^{(i_2)}_{\nu_1^2 \nu_2^2, r^2}$ by $\Delta$ steps up until its terminating local operator is aligned with the initial local operator of the other operator string $\hat{h}^{(i_2)}_{\nu_1^1 \nu_2^1, r^1}$ at step $\Delta = r^1 + r^2$. (g) Resubstituted local operators at step $\Delta = r^1 + 1$ with $r^2 = r^1 + 1$. The created operator string is $\hat{o}_{\nu_1^2} \otimes \epsilon_{\nu_1^1 \nu_1^2} \hat{o}_{\nu_1^1} \otimes \hat{\mathrm{Id}} \otimes \cdots \otimes \hat{\mathrm{Id}} \otimes \epsilon_{\nu_2^1 \nu_1^2} f^{r^1}_{\nu_1^1 \nu_2^1} \hat{o}_{\nu_2^1} f^{r^2}_{\nu_1^2 \nu_2^2} \hat{o}_{\nu_2^2}$, which is generated by the graph below.

The first lattice-ordered contribution is given via $\hat{H}_A$. The corresponding diagram is a single-branch graph obtained by identifying the final state of the graph $\Lambda(\hat{H}_{\nu^1, r^1})$ with the

initial state of the graph $\Lambda(\hat{H}_{v^2,r^2})$ by introducing an intermediate state $E$ (see fig. 12a). We now make use of the wedge product $\wedge$ for single-branched graphs as defined in eq. (30) to rewrite $\hat{H}_A$ in short as

$$\Lambda(\hat{H}_A) = \Lambda(\hat{H}_{r^1,v^1}) \wedge \Lambda(\hat{H}_{r^2,v^2}). \tag{52}$$

Swapping the operators, we obtain another lattice-ordered sum by commuting all local operator contributions, so that the corresponding graph picks up two factors, $\epsilon_{v_1^1 v_1^2} \epsilon_{v_2^1 v_2^1}$ for commuting $\hat{o}_{v_1^2}^{(i_2)}$ and $\epsilon_{v_1^1 v_2^2} \epsilon_{v_2^1 v_2^2}$ for commuting $\hat{o}_{v_2^2}^{(i_2+r^2)}$ with $\hat{o}_{v_1^1}^{(i_1)} \hat{o}_{v_2^1}^{(i_1+r^1)}$ (see fig. 12b). Again, the corresponding graph can be expressed via a wedge product,

$$\Lambda(\hat{H}_C) = \Lambda(\mathrm{sgn}(\hat{h}_{v_1^1 v_2^1,r^1}, \hat{h}_{v_1^2 v_2^2,r^2}) \hat{H}_{r^2,v^2}) \wedge \Lambda(\hat{H}_{r^1,v^1}), \tag{53}$$

where we have introduced $\mathrm{sgn}(\hat{h}_{v_1^1 v_2^1,r^1}, \hat{h}_{v_1^2 v_2^2,r^2}) \equiv \epsilon_{v_1^1 v_1^2} \epsilon_{v_2^1 v_1^2} \epsilon_{v_1^1 v_2^2} \epsilon_{v_2^1 v_2^2}$ for brevity. Note that, for $U(1)$-invariant operators, the identity $\mathrm{sgn}(\hat{h}_{v_1^1 v_2^1,r^1}, \hat{h}_{v_1^2 v_2^2,r^2}) \equiv 1$ holds.

The remaining sums over $\hat{h}_B(i_1, i_{i_2})$ correspond to overlapping interaction terms, i.e., all those lattice indices $i_1, i_2$ that fulfill

$$\{i_1, \cdots, i_1 + r^1\} \cap \{i_2, \cdots, i_2 + r^2\} \neq \emptyset. \tag{54}$$

In order to generate all these terms using one algorithm, we write the 2-point operator expressions graphically by representing a local operator $\hat{o}_v$ with a cross ("×") and the intermediate vacant operator sites with a circle ("○"), e.g.,

$$\hat{o}_{v_1}^{(i)} \hat{o}_{v_2}^{(i+r)} \rightarrow \times\!\!-\!\!\underbrace{\circ\!\!-\!\!\cdots\!\!-\!\!\circ}_{r-1 \text{ times}}\!\!-\!\!\times. \tag{55}$$

Employing this condensed notation, all lattice-ordered combinations of local operator strings can be generated by placing the graphical representations of $\hat{h}_{v_1^1 v_2^1,r^1}^{(i_1)}$ and $\hat{h}_{v_1^2 v_2^2,r^2}^{(i_2)}$ next to each other by aligning the last operator of $\hat{h}_{v_1^1 v_2^1,r^1}^{(i_1)}$ with the first operator of $\hat{h}_{v_1^2 v_2^2,r^2}^{(i_2)}$ (see fig. 12c). Subsequently, the right string is shifted upward until the initial operator of $\hat{h}_{v_1^1 v_2^1,r^1}^{(i_1)}$ is aligned with the final operator of $\hat{h}_{v_1^2 v_2^2,r^2}^{(i_2)}$ (see fig. 12f). While the right string's position is shifted by one step, the local operators $\hat{o}_{v_j^1}$ of the left operator string pick up a sign factor whenever they pass a local operator $\hat{o}_{v_i^2}$ in the right string, which is denoted by adding a factor $\epsilon_{v_i^1 v_j^2}$ to the left condensed representation (see figs. 12d and 12e).

For each such step $\Delta$, with $0 \leq \Delta \leq r^1 + r^2$, this algorithm generates a string $\hat{\gamma}_\Delta$ of local operators by merging the shifted right string into the left and resubstituting the original operators: $\times \rightarrow \hat{o}_v$. Missing sites are replaced with identities, whereas two local operators per site are contracted into a new local site operator $\hat{u}_{v_{12}} = \hat{o}_{v^1} \hat{o}_{v^2}$. Note that, in the latter case, the local operators are ordered in a way that avoids evaluation of on-site commutators $\left[\hat{o}_{v_i^1}^{(k)}, \hat{o}_{v_j^2}^{(k)}\right]$. Finally, each string $\hat{\gamma}_\Delta$ is converted into a single-branch graph by introducing a set of states $\left\{A_i^\Delta\right\}_i$ with the transitions between the $A_i^\Delta$'s properly chosen from the corresponding new local site operators $\left\{\hat{u}_{v_{12}}\right\}$ (see fig. 12g for an example).

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
