# Peer review of "Automated construction of $U(1)$-invariant matrix-product operators from graph representations"

_SciPost Physics, doi:SciPost Phys. 3, 035 (2017)_

## Round 1 · Referee Report · Anonymous (Referee 1) · 2017-7-31

Strengths

1. The method developed by the authors appears to be a useful insight that may well serve to produce more efficient algorithms for studying strongly correlated systems.
2. Meaningful example implementations of the method are included.
3. Evidence of algorithmic efficiency is provided.
4. Evidence is provided of tangible benefit when calculating an important quantity: the variance of the Hamiltonian (for a spin-1 chain).

Weaknesses

1. Confusing notation.
2. Very limited explanation of finite state machines (FSM), making the paper less self contained for non-experts.
3. Poorly explained figures.
4. Limited discussion of the work's context, relative to other recent MPO construction works.
5. No discussion of how this work may, or may not, be relevant to higher dimensional tensor methods (PEPS etc.).
6. Lack of equation numbers in many places is unhelpful to the reader.

Report

Matrix product state (MPS) algorithms for studying strongly correlated quantum systems in low dimensions rely heavily on efficient construction and representation of operators (such as the Hamiltonian) in the form of matrix product operators (MPOs).
One way to construct such representations is through the use of finite state machines (FSMs), however these can become very complicated themselves and somewhat unwieldy.

This manuscript seeks to address the issue of constructing generic, $U(1)$-invariant, matrix product operators using finite state machines. The authors exploit the language of graphs to simplify the translation of an FSM into an MPO, avoiding direct (and numerically expensive) MPO arithmetic. As an example they describe a concrete application, the variance of the Hamiltonian $\langle (H-E)^2 \rangle$ which is an important quantity for judging the convergence of MPS eigensolvers.

I think the work is sound and that the results are interesting and will be of use to the MPS community. However there are several issues that I believe the authors should attend to before publication.

I believe the authors should be more careful with their notation. For example, works on tensor methods are often unavoidably awash in a sea of indices, however I found tracking the meaning of indices in this manuscript more difficult than usual.
Some of this may be due to typos. In the following I give a list of places where I had issues with the notation.

a) On page 5, $\tau_j$ is the (right) block index for a tensor on site $j$, but on page 6, the $j$ in $\tau_j$ refers to the number of shift tensors applied (equivalently the number of two site gates applied). It might be useful to use some method, e.g. a tilde on the $j$ to discriminate between such cases.

b) In the caption of Fig. 2. the operators have lower indices (i.e. $\hat{A}_{(i)},\hat{B}_{(j)}$) for the site they act on, rather than the upper indices used in the rest of the text.

c) In Eqs. (2) and (3) the authors define $A^{n_i}$ and $B^{n_j}$, but I do not see these definitions used anywhere?

d) In the first paragraph of Sec. 3 just below the (unlabelled) display equation, the equation $\hat{h}_\nu^{(i)}=\hat{o}_{\nu_1}^{(i)}\cdots\hat{o}_{\nu_k}^{(i+k)}$ appears. The form of the first term in the product is inconsistent with the last, (substitute $k=1$ to see why) but the meaning of $k$ is not explained. I believe this is supposed to represent a string of $k$, possibly different, operators acting on sites $i$ through to $i+k$. I suspect the last term should be of the form
$\hat{o}_{\nu_p}^{(i+k)}$ where $p$ is independent of $k$ as they label separate things (a specific local operator from a set and the physical site respectively), as in Eq. 5.

e) In Eq. 5, the last equality defines an object $\hat{h}^r_{\nu_1,\cdots,\nu_n}(i)$. This changes the convention for $h$ used previously where $i$ was a superscript. Surely $\hat{h}^{(i)}_{\nu_1,\cdots,\nu_n}(r)$ or $\hat{h}^{(i)(r)}_{\nu_1,\cdots,\nu_n}$ would be better?

f) In Sec. 4 the superscript notation $\nu^1$ and $r^1$ is used. I think the superscript (rather than subscript) means that $\nu^1$ labels a particular set of operators rather than the first element of the set $\nu$ (which would be $\nu_1$?), but this is not explained.

g) The (non-symmetrized) wedge product $\wedge$ is used but not defined in the (unlabelled) equation between (7) and (8). The reader has to refer to the appendix to discover its definition.

This work hinges on understanding the operation of finite state machines. As such, and to make the work more self contained and useful to those less versed in FSM, I think the authors should sketch how to generate the $H_{XX}$ hamiltonian from a FSM, either in Sec. 3 or the caption of Fig. 3. In particular it is useful to explain the red and green color scheme, the meaning of branching etc.

The authors should do more to place their work in context with other recent work on efficient construction of generic MPOs, especially Hubig \emph{et al}. PRB 95, 035129 (2017). In that work the authors also deal with the issues of phase factors due to fermion anti-commutation and bookkeeping of auxiliary indices for conservation of quantum numbers. I believe the method presented here complements the work described in Hubig \emph{et al.} rather than replaces it, but would appreciate the authors' insights.
Furthermore in Sec. 5 and Fig 9. of this manuscript the authors present results for the variance of the Hamiltonian of a spin-1 chain of 100 sites. The same problem is also discussed by Hubig \emph{et al.} in their Sec. VIII and Fig. 10. I might expect these results to coincide, at least up to the bond dimension, $\chi \lesssim 100$, at which catastrophic cancellation becomes a problem. However they appear not to: can the authors explain this discrepancy? The results of Hubig et al. appear to be better for $\chi \lesssim 100$, even for the case with catastrophic cancellation.

Also in Sec. 5, it would be useful to know if the chain had open or periodic boundary conditions, and if the ground state was obtained using an iterative (i.e. sweeping) MPS method such as DMRG rather than "a standard Lanczos algorithm". At the end of the second paragraph of that section the authors refer to "perfect linear scaling", but the choice of axes in Fig. 8b makes this far from clear.

In the conclusion the authors refer to their algorithm as "optimized". It might be worth saying what quantity is extremal here?

Finally there are some other typos:

1. First paragraph of Sec 1., 2nd sentence. "developement" -> "development"
2. Fig 4. I think the rightmost graph is missing ellipses between its lines to indicate the multiple branches, as in the other two diagrams?
3. 2nd equation on page 8. This is not a spin flip term. Presumably "\hat{S}_{(i=2)}^+" should be "\hat{S}_{(i=2)}^-"?
4. Page 9, 2nd sentence, "are acting" -> "act".
5. Page 11, Fig 8a is referenced before Fig 7.
6. Page 12, just below Eq. 10, the sentence "In the case of the variance" is very hard to parse and not grammatically correct. I suggest, "we obtain in double ..." -> "with double precision arithmetics, $p=p_{num}=16$, the n\"{a}ive evaluation, $\delta<16-2\gamma$, yields a maximal possible precision of ...".
7. Last sentence of the conclusion: "from increasing numerical precision and gain" -> "from the increased numerical precision and gain".
8. Page 18, second from last equation. I don't think the ellipsis should appear on the 2nd line, and there should be a leading $+$ on the 3rd line.
9. Fig. 11 caption $\hat{h}_b$ -> $\hat{h}_B$.

Requested changes

1. Improve notation for clarity.
2. Describe how a finite state machine is used to generate e.g. a Hamiltonian, perhaps in the caption of Fig 3. This would also be the correct place to describe what the green and red colors mean in terms of the FSM!
3. Explain how this work (especially the discussion in Sec. 5 and of Fig. 9.) complements that by Hubig, McCulloch and Schollwock PRB 95, 035129 (2017), especially with ref to the results in Fig. 9.
4. Number the equations to make things clearer for the interested reader (and referee) who wishes to refer back to different parts of the manuscript.
5. Define $\wedge$ product when it first appears rather than waiting until the appendix.
6. Improve Fig. 8. Panel a is referenced before Fig. 7 and panel b does not make the linear scaling clear.
7. Fix the other numbered typos described in the report.

  • validity: high
  • significance: good
  • originality: good
  • clarity: ok
  • formatting: good
  • grammar: excellent

Author:  Sebastian Paeckel  on 2017-09-28  [id 176]

(in reply to Report 1 on 2017-07-31)

We thank the referee for his or her careful reading of the manuscript and for the positive assessment "the work is sound and that the results are interesting and will be of use to the MPS community." In the following, we address the issues raised. We have uploaded an updated version including these changes to the arXiv, where the manuscript is accessible under the identifier xxxv2.

Referee:
"I believe the authors should be more careful with their notation. For example, works on tensor methods are often unavoidably awash in a sea of indices, however I found tracking the meaning of indices in this manuscript more difficult than usual.
Some of this may be due to typos. In the following I give a list of places where I had issues with the notation."

We thank the referee for this remark. We have carefully taken care of the various points concerning notation and removed some typos. We hope the manuscript is clearer now.

"This work hinges on understanding the operation of finite state machines...."

We have now included a description of finite state machines and sketch the construction for a specific example (Hamiltonian of the XX model).

"The authors should do more to place their work in context with other recent work on efficient construction of generic MPOs, especially Hubig \emph{et al}. PRB 95, 035129 (2017). "

We now explain in more detail how our work relates to prior work in the end of section "Numerical behavior of the variance for antiferromagnetic $S=1$ Heisenberg chains". We state that the approach of Hubig et al is complementary to ours. The additional advantages of our FSM approach is that the MPO is generated from an abstract graph operation, so that the representation is exact, leading to the higher accuracy.
Note that to our understanding, Hubig et al. make use of non-abelean symmetries in their paper, so that it is not possible to directly compare to the bond dimensions used by us.

"Also in Sec. 5, it would be useful to know if the chain had open or periodic boundary conditions, and if the ground state was obtained using an iterative (i.e. sweeping) MPS method such as DMRG rather than "a standard Lanczos algorithm". At the end of the second paragraph of that section the authors refer to "perfect linear scaling", but the choice of axes in Fig. 8b makes this far from clear."

We apply open boundary conditions, and the ground state is obtained by sweeping through the lattice and locally optimizing the site tensors using a Lanczos algorithm. Both is now stated in the text.
We now also show linear fits in Fig. 8b, which clearly demonstrate the linear scaling.

"In the conclusion the authors refer to their algorithm as "optimized". It might be worth saying what quantity is extremal here?"
In the main text we now address this point - the 'optimization' refers to minimizing the number of floating point operations when calculating expectation values of operator products, as needed in the evaluation of the variance.

"Finally there are some other typos:"
We have corrected typos throughout the manuscript.

Requested changes:
All requested changes are included.

---

## Round 2 · Referee Report · Anonymous · 2017-10-19

Report

I think the changes made in response to my previous report are satisfactory, particularly the discussion at the end of section 5 on the context of this work in relation to that of Hubig et al., and I recommend publication.
There are two small typos.

Requested changes

1. 3rd paragraph of the introduction: "capable to efficiently" -> "capable of efficiently".
2. Footnote 8, "non-abelean" -> "non-abelian".

---

## Round 2 · Author Response

We thank the referee for his or her careful reading of the manuscript and for the positive assessment "the work is sound and that the results are interesting and will be of use to the MPS community." In the following, we address the issues raised. We have uploaded an updated version including these changes to the arXiv, where the manuscript is accessible under the identifier 1706.05338v2.

Referee:
"I believe the authors should be more careful with their notation. For example, works on tensor methods are often unavoidably awash in a sea of indices, however I found tracking the meaning of indices in this manuscript more difficult than usual. Some of this may be due to typos. In the following I give a list of places where I had issues with the notation."

We thank the referee for this remark. We have carefully taken care of the various points concerning notation and removed some typos. We hope the manuscript is clearer now.

"This work hinges on understanding the operation of finite state machines...."

We have now included a description of finite state machines and sketch the construction for a specific example (Hamiltonian of the XX model).

"The authors should do more to place their work in context with other recent work on efficient construction of generic MPOs, especially Hubig \emph{et al}. PRB 95, 035129 (2017). "

We now explain in more detail how our work relates to prior work in the end of section "Numerical behavior of the variance for antiferromagnetic S=1 Heisenberg chains". We state that the approach of Hubig et al is complementary to ours. The additional advantages of our FSM approach is that the MPO is generated from an abstract graph operation, so that the representation is exact, leading to the higher accuracy.
Note that to our understanding, Hubig et al. make use of non-abelean symmetries in their paper, so that it is not possible to directly compare to the bond dimensions used by us.

"Also in Sec. 5, it would be useful to know if the chain had open or periodic boundary conditions, and if the ground state was obtained using an iterative (i.e. sweeping) MPS method such as DMRG rather than "a standard Lanczos algorithm". At the end of the second paragraph of that section the authors refer to "perfect linear scaling", but the choice of axes in Fig. 8b makes this far from clear."

We apply open boundary conditions, and the ground state is obtained by sweeping through the lattice and locally optimizing the site tensors using a Lanczos algorithm. Both is now stated in the text.
We now also show linear fits in Fig. 8b, which clearly demonstrate the linear scaling.

"In the conclusion the authors refer to their algorithm as "optimized". It might be worth saying what quantity is extremal here?"
In the main text we now address this point - the 'optimization' refers to minimizing the number of floating point operations when calculating expectation values of operator products, as needed in the evaluation of the variance.

"Finally there are some other typos:"
We have corrected typos throughout the manuscript.

Requested changes:
All requested changes are included.

---

## Round 2 · List of Changes

We have included several changes throughout the paper in reply to the referee's request, as detailed in our reply above. In more detail, the major changes are:

- We have added numbers to the equations, unified the notation of the tensors, and corrected typos throughout the paper.

- Figs. 1 and 2 have been updated for clarity

- Sec. 3.1 / Fig. 3 provide the example of constructing a FSM for the XX model

- We modified the discussion in Sec. 5 to better relate to the work of Hubig et al.

You are currently on this page

Resubmission 1706.05338v2 on 6 October 2017

---

## Editorial Decision

published